# Tunable Thermal Transport Characteristics of Nanocomposites

**DOI:** 10.3390/nano10040673

**Published:** 2020-04-03

**Authors:** G. P. Srivastava, Iorwerth O. Thomas

**Affiliations:** School of Physics, University of Exeter, Stocker Road, Exeter EX4 4QL, UK; i.o.thomas2@exeter.ac.uk

**Keywords:** thermal transport, phonons, nanocomposites, DFT, Boltzmann equation, effective medium theory

## Abstract

We present a study of tunable thermal transport characteristics of nanocomposites by employing a combination of a full-scale semi-*ab inito* approach and a generalised and extended modification of the effective medium theory. Investigations are made for planar superlattices (PSLs) and nanodot superlattices (NDSLs) constructed from isotropic conductivity covalent materials Si and Ge, and NDSLs constructed from anisotropic conductivity covalent-van der Waals materials MoS${}_{2}$ and WS${}_{2}$. It is found that difference in the conductivities of individual materials, period size, volume fraction of insertion, and atomic-level interface quality are the four main parameters to control phonon transport in nanocomposite structures. It is argued that the relative importance of these parameters is system dependent. The equal-layer thickness Si/Ge PSL shows a minimum in the room temperature conductivity for the period size of around 4 nm, and with a moderate amount of interface mass smudging this value lies below the conductivity of SiGe alloy.

## 1. Introduction

Thermal conductivity is a property of bulk solids spanning over four orders of magnitude, covering the range $10^{-1}$–$10^{3}$ W m${}^{-1}$ K${}^{-1}$ [1] at room temperature. This range can be further increased by including solids of nanoscale size [1] and even more so through also considering nanocomposites. Both high and low thermal conductivity materials have technological applications. Materials with high thermal conductivities can be used as sinks for managing heat in electronic devices [2]. Materials with ultra-low thermal conductivities are suitable for thermal barrier coatings for aircraft and gas-turbine engines [3] and for achieving high thermoelectric figure of merit [4,5]. Theoretical work by Hicks and Dresselhaus [6,7] provided proof of the concept that nanostructing, resulting in lowering of thermal conductivity, greatly enhances the thermoelectric figure of merit. In a review article Dresselhaus et al. [8] presented the status of experimental and theoretical works in the emerging field of low-dimensional thermoelectricity, and discussed the outlook for future research directions for nanocomposite thermoelectric materials. However, in their theoretical works, Hicks and Dresselhaus did not present any numerical calculations of phonon conductivities for the nanostructures they had studied. Hence, the identification of key physical parameters of nanostructures, particularly nanocomposites, for tuning phonon transport remains an important topic of both fundamental and practical importance.

A systematic theoretical study of thermal conduction across interfaces in composite materials ranging from short (nanometers) to large (micrometers or beyond) periodicities is quite challenging. Indeed, no such study as far as we are aware has yet been performed. A few first-principles-based attempts have been made, but due to heavy computational demands are restricted to period sizes in the low nanometer (nm) range [9,10,11]. In this work we employ a combination of the electronic density functional theory (DFT) and the phonon Boltzmann transport equation based semi-ab initio approach and a generalised and extended modification of effective medium approach (GemEMA) to study tunable thermal transport characteristics of composites of periodicities ranging from 1 nm to 100 μm, with particular emphasis on nanocomposites of periodicities in the range 1–100 nm.

## 2. Theoretical Framework

At the start we present in Table 1 a list of the important quantities and acronyms used in this work.

Thermal transport calculations based on phonon relaxation times assume the validity of the Landau-Peierls-Ziman condition [12]: sample size (*L*) must be larger than the phonon mean free path (Λ), which in turn must be larger than the phonon wavelength (λ), i.e., L>Λ>λ. Figure 1 illustrates a planar superlattice (PSL) structure of sample size *L* and repeat period size *D* containing layers of materials A and B. Also shown in that figure is a nanodot superlattice (NDSL) structure with dot A inserted in matrix B. The most straightforward solution of the Boltzmann transport equation gives an expression for the phonon conductivity containing a simple form of phonon relaxation time, called the single-mode relaxation time (SMRT) [13]. Callaway [14] developed the concept of an effective relaxation time by incorporating the momentum conservation condition of anharmonic Normal scattering processes. However, the original Callaway conductivity expression, which has been applied extensively in early [15,16,17] as well as recent [18] works, employs grossly simplified and parameterized forms of anharmonic relaxation times, is valid for isotropic materials and is based on the continuum approximation to the phonon dispersion relation $\omega \left(qs\right)=c_{ac}\left(s\right)q$, where $c_{ac}$, *s* and q are acoustic speed, polarization and wavevector, respectively. We have recently presented [19] derivation of a generalised version of Callaway’s formalism (i) accounting for the tensorial form of thermal conductivity and (ii) utilizing realistic phonon dispersion relations ω=ω(qs) for all polarization branches.

We can establish the range of validity of relaxation-time based full-scale and effective medium theories of thermal transport across interfaces in a composite system by comparing the phonon mean free path (Λ) with the repeat period size (*D*) of the system under study. Three regions can be considered: D<<Λ, D∼Λ and D>>Λ, as illustrated in Figure 2. For the case D<<Λ the system is characterised by a unique set of phonon dispersion relations ω(qs) throughout the relevant Brillouin zone. In such cases full phonon eigensolutions must be obtained and a full-scale conductivity calculation carried out. Such calculations for systems with D∼Λ may be too computationally demanding to be viable, and so the application of approximate methods, such as an appropriate effective medium approach, becomes necessary. In applying such an approach the necessary inputs are the phonon conductivities of constituent bulk materials and the thermal interface (boundary) resistance $R_{\mathrm{TB}}$. For D>>Λ the effect of boundary in a composite structure becomes less important and the thermal resistivity of the system can be obtained from a suitable linear combination of the resistivities of constituent bulk materials. For a two-component composite A/B with with repeat period *D* the thermal conductivity κ for the three cases can thus be viewed as follows: (1)$$\begin{array}{ccc} \mathrm{Region}\;R1\;(D<<\Lambda ): & \kappa = & \kappa (\mathrm{system}\;\mathrm{considered}\;\mathrm{as}\;a\;\mathrm{new}\;\mathrm{material}), \end{array}$$(2)$$\begin{array}{ccc} \mathrm{Region}\;R2\;(D∼\Lambda ):\; & \kappa = & \kappa (a\;\mathrm{suitable}\;\mathrm{effective}\;\mathrm{medium}\;\mathrm{expression}), \end{array}$$(3)$$\begin{array}{ccc} \mathrm{Region}\;R3\;(D>>\Lambda ): & \kappa = & a\;\mathrm{suitable}\;\mathrm{combination}\;\mathrm{of}\;\mathrm{component}\;\mathrm{bulk}\;\mathrm{conductivities}. \end{array}$$

For *ultra short period* nanocomposites, we calculate κ(D<<Λ) using our semi-ab initio approach for the DFT-Boltzmann based conductivity tensor [11,20] and a generalised version of Callaway’s anharmonic phonon relaxation time expression [19] using phonon eigensolutions generated using Density Funtional Peturbation Theory [21]. A recap of the essential features is presented in Appendix A. For composites with repeat period sizes in large nm and μm ranges, we calculate κ(D≥Λ) (particularly for D∼Λ)) by applying the effective medium approach at three levels. (a) In Nan’s approach [22] $\kappa =\kappa (L,f,R_{TB})$ is calculated by assuming sharp interfaces and inputting $\kappa_{A}\left(L\right)$ and $\kappa_{B}\left(L\right)$ for the composite sample size *L* and volume fraction *f* of A in B. (b) In the Minnich-Chen approach [23] $\kappa =\kappa (L_{I},L_{M},f,\Sigma ,R_{TB})$ is calculated by inputting $\kappa_{I}\left(L_{I}\right)$ and $\kappa_{M}\left(L_{M};L_{\mathrm{IS}}\right)$, where $L_{I}$ is effective length of insertion A, $L_{M}$ is effective length of matrix B, and $L_{\mathrm{IS}}$ is the effective scattering length arising from the density of interfaces Σ. (c) Our recent extension [24] of the Minnich-Chen theory in (b) includes the effects of anisotropy within the composite matrix and the anisotropy of the thermal boundary resistance $R_{TB}$. (d) Another extension of Chen’s theory incorporates non-uniformity of interfaces in evaluating $R_{TB}$ [25,26]. (e) In this work we extend the theory further by accounting for the effect of unavoidable presence of atomic defects at interfaces (such as mass smudging), *viz.* by making a realistic estimate for $W_{\mathrm{IMS}}$. This we do by considering a small percentage of mass swap in each constituent material that mimics the effects of swapping masses within the layers on either side of an interface, as discussed in more detail below. We will label conductivity expressions within Nan’s approach as $\kappa_{\mathrm{EMA}}$, within Chen’s approach as $\kappa_{\mathrm{mEMA}}$, incorporating considerations (b–d) as $\kappa_{\mathrm{emEMA}}$, and incorporating considerations (b–e) as $\kappa_{\mathrm{GemEMA}}$. The constituent components $\kappa_{I}$ and $\kappa_{M}$ are computed using our semi-ab initio approach from DFPT generated eigensolutions.

Within the GemEMA scheme the cross-plane conductivity expressions for planar superlattices (PSLs) and nanodot superlattices (NDSLs), relevant for RegionR2(D∼Λ), are:(4)$$\mathrm{Region}\;\;R2\;\left(D∼\Lambda \right):\;\kappa =\left\{\begin{array}{cc} \kappa^{M}\frac{\kappa^{I}}{\kappa^{I}-f\left[\kappa^{I}\left(1-\alpha \right)-\kappa^{M}\right]}, & \mathrm{PSL}\\ \kappa^{M}\frac{\kappa^{I}\left(1+2\alpha \right)+2\kappa^{M}+2f\left[\kappa^{I}\left(1-\alpha \right)-\kappa^{M}\right]}{\kappa^{I}\left(1+2\alpha \right)+2\kappa^{M}-f\left[\kappa^{I}\left(1-\alpha \right)-\kappa^{M}\right]}. & \mathrm{NDSL} \end{array}\right.$$

Here the superscript M represents the segment B as matrix of conductivity $\kappa^{M}$, the superscript I represents the segment A as insert with (a small) volume fraction *f* and conductivity $\kappa^{I}$, and the dimensionless parameter α is the ratio $\alpha =\kappa^{M}R_{\mathrm{TB}}/L_{B,I}$ with $R_{\mathrm{TB}}$ as the thermal boundary resistance (TBR) and $L_{B,I}$ as an effective insert boundary length. The parameter α is evaluated by incorporating in $R_{\mathrm{TB}}$ a momentum dependent inhomogeneity of the interface region [27]. With α=0 the interface is perfect and there is no thermal boundary resistance. Sihovola’s approach [28] for anisotropic matrix and insert has been extented to the interface region for calculating anisotropic $R_{\mathrm{TB}}$ [24]. We consider phonon scattering from sample boundary to be purely diffusive. But in order to account for a realistic nature of interfaces phonon boundary scattering is expressed as wave-vector dependent [25], following the phenomenological scheme proposed by Koh et al. [27]. This approach provides an improvement over the approach by Behrang et al. [29] for including both specular and diffuse contributions to insert and matrix boundary scattering rates as well as the thermal boundary resistance. The fraction of phonon modes of momentum q that undergo diffuse scattering from interfaces/boundaries is expressed as [27] $s_{q}=1-e^{\eta^{2}{\left|q\right|}^{2}}$ with $\eta =\epsilon /a_{0}$, where ϵ is the average height of surface inhomogeneity in units of the lattice spacing $a_{0}$. A physically meaningful value of η lies somewhere between 0 (the specular limit for a smooth surface) and *∞* (the diffuse limit for an infinitely rough surface). We have considered η=2 a resonable value in the present work. The effective boundary length $L_{B}$ is then expressed as
(5)$$\begin{array}{cc} \mathrm{PSLs}:\;L_{B,i}^{-1} & =s_{q}L_{i}^{-1}+\left(1-s_{q}\right)L^{-1}\;\mathrm{for}\;i^{\mathrm{th}}\mathrm{segment},\;i=I,M \end{array}$$
(6)$$\begin{array}{cc} \mathrm{NDSLs}:\;L_{B}^{-1} & =\left\{\begin{array}{c} L_{B,I}^{-1}=s_{q}L_{I}^{-1}+\left(1-s_{q}\right)L^{-1}\;\mathrm{for}\;\mathrm{inserts}\\ L_{B,M}^{-1}=L^{-1}+s_{q}L_{\mathrm{IS}}^{-1}\;\mathrm{for}\;\mathrm{the}\;\mathrm{matrix}, \end{array}\right., \end{array}$$
where *L* is the sample boundary length, and $L_{I}$ is the insert boundary length (nanodot diameter *d* for NDSLs), $L_{i}=\left(d_{A},d_{B}\right)$ for PSL segment (A,B) of periodic size $D=d_{A}+d_{B}$, and $L_{\mathrm{IS}}=4/\Sigma$ is the effective scattering length arising from the density of interfaces. For PSLs Σ=2/D and for NDSLs $\Sigma =6f/L_{I}=6f/d$. The TBR is then considered as the sum of weighted diffuse and specular constributions: $R_{\mathrm{TB}}=R_{\mathrm{TB}}^{\mathrm{Diff}}+R_{\mathrm{TB}}^{\mathrm{Spec}}$. Finally, the effect of including interface mass smudging (IMS) is to essentially generate an additional contribution to TBR: in the case of PSLs we can express $R_{\mathrm{TB}}^{\mathrm{eff}}=R_{\mathrm{TB}}+D\;W_{\mathrm{IMS}}$. (The presence of form factors in NDSLs results in a more complicated relationship.) It is reasonable to assume a small amount of mass swap emulating the effects of smudging involving just one or two atomic layers to compute the corresponding thermal resistivity $W_{\mathrm{IMS}}$. It is useful to mention that there is no restriction on the validity of the effective medium theories for PSLs, and that the methods are valid in NDSLs for small IF densities (typically for f≤0.25).

The GemEMA expressions in Equation (Equation 4) become the mEMA expressions of Ref. [23] for isotropic systems (*viz.* when $\kappa_{I}$, $\kappa_{M}$ and $R_{\mathrm{TB}}$ are non-tensorial) and when there is no cross-interface mass swapping and transmission across the interface is purely diffuse (*viz.* when η=∞ and $W_{\mathrm{IMS}}=0$). And the mEMA expressions reduce to Nan’s EMA expressions when $L_{B,I}=L_{B,M}=L$ (sample size).

In RegionR3(D>>Λ), when the parameter α becomes negligibly small, and $L_{B,I}=L_{B,M}=L$, the expressions in Equation (Equation 4) can be reduced to
(7)$$\mathrm{Region}\;\;R3\;\left(D>>\Lambda \right):\;\frac{1}{\kappa }=\left\{\begin{array}{cc} \frac{f}{\kappa^{I}}+\frac{1-f}{\kappa^{M}}, & \mathrm{PSL}\\ \frac{2+f}{\nu }\frac{1}{\kappa^{I}}+\frac{1-f}{\nu }\frac{1}{\kappa^{M}}, & \mathrm{NDSL} \end{array}\right.$$
where $\nu =\left(1+2f\right)+2\left(1-f\right)\kappa^{M}/\kappa^{I}$. This equation helps explain easily that, for a given insert fraction *f*, the larger the difference between the matrix and insert conductivities, the larger the change in conductivity of the composite compared to that of matrix: i.e., a larger value of $\left(\kappa^{M}/\kappa^{I}-1\right)$ results in a smaller value of $\kappa /\kappa^{M}$.

## 3. Methodology

### 3.1. Generation of Phonon Eigensolutions

Phonon eigensolutions required for the thermal conductivity calculations were generated using the Quantum Espresso package [21] with parameters defined in previous studies [11,20,30]; some quantitative corrections from previously reported thermal conductivity results are due to this study’s work using the correct eigenvalue input as per recent work [26].

### 3.2. Computation of Mass Defect and Anharmonic Scatterings

Phonon scattering rates from isotopic mass defects and crystal anharmonicity in bulk materials and for *ultra short period* nanocomposites were calculated using the procedure described in our previous publications [11,20].

### 3.3. Computation of IMS Scattering

The phonon relaxation rate due to interface mass smudging (IMS) in *ultra short period* nanocomposites is computed by employing the perturbative scheme of treating mass defects as explained in a previous work [11]. Specifically, we considered IF mass swapping using the Gaussian distribution of the type $\exp(-j^{2}\zeta )$ where ζ is a parameter determining the amount $j^{2}\zeta$ of mass swap in the *j*th interface layer. We have presented numerical results by considering mass swap between only one atomic layer across the interface. This is illustrated in Figure 3. For 5% and 10% mass swapp on the first layer (j=1) across an interface, we set ζ=3.5 and ζ=2.3, respectively. Extending the consideration to the second IF layer increases the relaxation rate by less than 1%. For composites with large repeat periods it is not possible to include the effect of IMS directly as discussed above for ultra short period nanocomposites. Instead, we have adopted an alternative, more simple scheme. Considering a unit cell of suitable shape and size for one of the bulk constituent materials we replace the relevant atom with the desired fraction of the atom swapped from the other constituent bulk material. The phonon relaxation rate is computed employing the same method as used for isotopic impurites in bulk materials [11,13,20]. Using this as input, thermal resistivity $W_{\mathrm{IMS}}$ is computed within the SMRT scheme. The overall conductivity for the constituent is then calculated by summing its conductivity with this resistivity via Mathiesen’s rule.

## 4. Results

### 4.1. Predicted Results for Modelled Periodic
Nanocomposite Structures

Figure 4 shows the variation of the room-temperature cross-plane κ with the repeat period *D* in Si/Ge PSLs and NDSLs, of sample boundary length 1 mm. Also presented are the in-plane and cross-plane κ results for MoS${}_{2}$/WS${}_{2}$ NDSLs. These results have been obtained by including isotopic mass defects, but otherwise with a homogeneous interface between the constituent layers. Regions R1, R2 and R3, corresponding to D<<Λ, D∼Λ and D>>Λ, respectively, have been indicated. Full-scale calculations for D<10 nm have been made by treating the nanocomposite systems as new materials. For D>10 nm results of calculations have been presented using the last expression in Equation (3) as well using Nan’s EMA [cf. Equation (2)]. It is obvious that Region R3 starts when *D* becomes larger than 10 microns. Region R2 may safely be considered for the range 10nm<D<1micron. The boundary between the regions R1 and R2 at around D∼10 nm has been chosen following the suggestion [1] that typical minimum for phonon mean free path is 10 nm. This can also be judged from our numerical results for Λ in Figure 5 discussed below. The cross-plane conductivity component $\kappa_{zz}$ for the Si/Ge PSL system decreases as *D* increases from an ultralow nm value, takes a minimum in the range 3–12 nm, and then continues to increase until 10 μm, before saturating to the bulk weighted result obtained from Equation (3).

The typical cross-plane conductivity shape in Figure 4a for the repeat period range 1–12 nm (sample size 1 mm) can be explained by considering two physical properties [9,31]: cross-plane phonon group velocity $c_{z}$ and interface density Σ. As seen in Figure 5, both $|c_{z}|$ and Σ decrease as *D* increases up to about 4 nm. For D> 4 nm, while Σ continues to decrease linearly with *D*, the decrease in $|c_{z}|$ is much reduced. Decrease in $|c_{z}|$ and Σ have opposite effects: the former reduces and the latter increases the cross-plane conductivity ($\kappa_{zz}$). The combined effects of changes in $|c_{z}|$ and Σ, together with increase in specific heat $C_{v}$, result in an increase in $\kappa_{zz}$ for D≥8 nm. Consistent with this consideration is the variation of the Bose occupation $\bar{n}\left(qs\right)$ weighted ’cross-plane mean free path’ $\Lambda_{z}=|c_{z}|\tau =\sum_{qs}\left|c_{z}\left(qs\right)\right|\tau \left(qs\right)\bar{n}\left(qs\right)/\sum_{qs}\bar{n}\left(qs\right)$, which shows a minimum at 4.4 nm in Figure 5. Notice that in contrast to $\Lambda_{z}$ the occupation weighted mean free path $\Lambda =|c|\tau =\sum_{qs}\left|c\left(qs\right)\right|\tau \left(qs\right)\bar{n}\left(qs\right)/\sum_{qs}\bar{n}\left(qs\right)$ does not show any dip as *D* increases.

For technological applications of nanocomposites, relevant considerations are the sample size in 100s of nm and the repeat period in the range 1–50 nm. There are four fabrication-related parameters whose role in limiting cross-plane phonon transport must be examined systemtically. These are: the material identity of the insert and matrix components, repeat period size, volume fraction of the insert component in a period (relating to interface density), and the quality of fabricated nanocomposite relating to interface homogeneity and the inevitable presence of defects (point defects such as mass smudging, and more complicated varieties such as dislocations and grain boundaries). In this work we have considered the role of interface mass smudging (IMS) using our semi-ab initio method as applied to bulk materials, and have restricted ourselves to the consideration of inhomogeneity in a phenomenological manner as mentioned earlier. Before examing the role of these fabrication-related parameters, we clarify that phonon conductivity of any material, be it pure bulk or composite, varies with sample length. From our theoretical calculations we estimate that for a sample size of 500 nm the room-temperature conductivity results for bulk Si and Ge are 81.62 and (37.07, 52.17) (natural, enriched) W/m/K, respectively. For the same sample size (500 nm) the (in-plane, cross-plane) room-temperature conductivity results for 2H bulk MoS${}_{2}$ and WS${}_{2}$ are (84.28,2.58) and (108.20,1.75) W/m/K, respectively.

The results in Figure 6a suggest that for Si/Ge PSL of sample size 1 mm $\kappa_{\mathrm{mEMA}}$ becomes almost the same as $\kappa_{\mathrm{EMA}}$ when the period size grows up to several tens of micron. Clearly, mEMA is more appropriate for composites with period size ranging between nanometers and a few microns. Figure 6b shows a comparison of the results from the emEMA method with those from the mEMA and EMA methods for a Si/Ge PSL of sample size 500 nm. Using a reasonable choice of η=2 for interface inhomogeneity, $\kappa_{\mathrm{emEMA}}$ lies in between $\kappa_{\mathrm{EMA}}$ and $\kappa_{\mathrm{mEMA}}$. From the results presented in panel (c) three points can be made. Increasing the insert volume fraction of Ge in a Si matrix lowers the conductivity of the nanocomposite. For a given insert fraction, the decrease in the conductivity is more pronounced even for lower amounts of interface mass smudging. Finally, even a small amount of IMS can alter κ more strongly than adding a larger volume of insert. For example, it is clear that 10% of mass smudging (ζ=2.3 for j=1, see Methodology section) across the first atomic layers of Si and Ge lowers the conductivity more than what does doubling of the Ge insert volume fraction. Consideration of an additional 0.01% mass smudging across the second interface atomic layers (ζ=2.3 for j=2) reduces the conductivity by another 0.06% (not shown in the figure).

Figure 7 presents the main results of our investigation for room-temperature cross-interface conductivity with sample boundary length fixed at 500 nm. Our numerical results for the thermal boundary resistance are: $R_{\mathrm{TB}}=2.392\times 10^{-9}$ m${}^{2}$ K W${}^{-1}$ for Si/Ge, and $R_{\mathrm{TB}}^{\mathrm{in}-\mathrm{plane}}=3.211\times 10^{-9}$ m${}^{2}$ K W${}^{-1}$ and $R_{\mathrm{TB}}^{\mathrm{cross}-\mathrm{plane}}=5.717\times 10^{-8}$ m${}^{2}$ K W${}^{-1}$ for MoS${}_{2}$/WS${}_{2}$. In presenting results in panel (a) for Si(D/2)/Ge(D/2) PSLs we have combined the numerical data obtained from the full-scale calculations for period sizes 1.1–9.9 nm and from GemEMA (with ζ=2.3) for larger period sizes. In panel (b) we have shown results for the Si/Ge NDSLs with Ge insertion volume fraction 0.125 in Si matrix using the full-scale calculation for D=1.1 nm and GemEMA (ζ=2.3) for larger *D* values. Looking at panels (a), (b) and (c) it is evident that as the period size increases beyond 10 or 15 nm so does the cross-interface conductivity of a nanocomposite, be it a PSL of a NDSL. The results in panel (a) reveal that the conductivity of the Si/Ge PSL can be expected to acquire a minimum value when the period size lies in the range D= 3–12 nm, before starting to increase for lower values of *D*. The effect of including $W_{\mathrm{IMS}}$ to account for mass smuding across interfaces (IMS) is to lower the value of the conductivity. The extent to which IMS reduces conductivity depends on the atomic masses involved at the interface layer(s). For Si/Ge(001) and Si/Ge(111) systems there is a single atomic layer of each species across an interface. For Si/Ge(110) there would be two atoms of each species across each interface. For composites made of transition metal dichalogenides such as MoS${}_{2}$/WS${}_{2}$(0001), with each layer containing one transition metal element and two atoms of a chalcogen element, only a maximum of 1/3 atomic site can be swapped across the first interface layer. Consistent with this, and that interlayer separation in these van der Waals materials is quite large, our results in panel (c) suggest that even with a 10% atomic swap between Mo and W for MoS${}_{2}$/WS${}_{2}$ NDSLs there is only a minimal decrease in the conductivity. Another variable structural parameter is the volume fraction of insertion. In general, if there is a large difference in the conductivities of insert and matrix bulk materials, there will be a bigger difference in the conductivity of a nanocomposite when the volume fraction of the insert increases. This was found to be the case for both Si/Ge and MoS${}_{2}$/WS${}_{2}$ systems. As shown in panel (c) we find that the conductivity of MoS${}_{2}$/WS${}_{2}$ NDSL decreases roughly by a factor of 1.5 when the volume fraction of the insert WS${}_{2}$ doubles from 0.125 to 0.25.

From an inspection of the results obtained for the Si/Ge and MoS${}_{2}$/WS${}_{2}$ nanocomposites, it is clear that both the rate of increase in conductivity with period size *D* and decrease in the conductivity with insert volume fraction *f* can be affected by the amount of IMS present. However, this is dependent on the chemical and structural makes of the insert and matrix. While the effect of IMS is stronger in Si/Ge NDSLs and PSLs, it is minimal in MoS${}_{2}$/WS${}_{2}$ NDSLs. With the consideration of 10% first-layer IMS for a sample of length 500 nm, we predict the minimum conductivity value of around 2 Wm${}^{-1}$K${}^{-1}$ for the Si(4.4 nm)/Ge(4.4 nm) PSL and around 3.4 Wm${}^{-1}$K${}^{-1}$ for a NDSL with Ge nanodot of 0.56 nm diameter and period size 1.1 nm.

### 4.2. Theory-Experiment Comparison

Our predicted values of the room-temperature thermal conductivity for Si/Ge nanocomposites are in the same range as reported from experimental measurements on similar size PSLs [32,33,34,35] and NDSLs [36,37,38,39]. However, it should be clarified that a detailed comparison of our predicted results for Si/Ge nanocomposite structure with experimental measurements is not possible. This because in our theoretical studies we have assumed samples to be homogeneous with perfectly periodic structure at atomic level, except for some degree of mass smudging at interfaces. In contrast, even the best fabrication techniques result in samples characterised with inhomogeneities and a large number of point and extended defects. Having said that, it is tempting to make a detailed comparison of our predicted results for equal-layer short-period Si/Ge PSLs of sample size (*L*) of around 400 nm and period size (*D*) of 4.4 nm with the reported experimental results in the temperature range 50–300 K as reported in Ref. [33]. Figure 8a shows our theoretical results using the SMRT and Callaway expressions for the conductivity of the Si(2.2 nm)/Ge(2.2 nm) PSL of sample size 400 nm n-doped with 10${}^{26}$ m${}^{-3}$, and the IMS factor ζ=2.3. We used Parrott’s expression [40] for phonon scattering rate from donors. The results, throughout the temperature range, from the Callaway theory are slightly higher than those from the SMRT theory. At 300 K, κ(Callaway) is 3% larger than κ(SMRT). The theoretically obtained room-temperature result in this figure is very close to the experimental result in Ref. [33]. However, there is huge descrepancy between theoretical and experimental results below room temperature. As mentioned before, this is due to several known factors related to the quality of the fabricated sample at chemical and atomic levels. We made calculations by doubling the mass defect factor Γ(md) and using a slightly smaller effective sample size to examine if this would help our results closer to experimental results. It is clear from Figure 8b that a simple scaling of mass defect concentration is not sufficient to explain experimental results. Indeed, the plateau-like feature in the experimental curve in the range 50–250 K is indicative of gross inhomogeneity or amorphousness. We believe that had the measurements been made for temperatures above 300 K where anharmonic phonon interactions play a dominant role, our theoretical curve would have matched with experimental curve. Clearly, more work is needed both on experimental side (on fabrication and quality assessment at atomic and chemical levels) and theoretical side to fully establish precise numerical values of the conductivity at temperatures below 300 K. The same argument stands when comparing our theoretical results with measurements for unequal-layer Si/Ge PSLs reported in Ref. [34]. It is also pleasing to note that our computed value of the Si/Ge thermal boundary resistance $R_{\mathrm{TB}}=2.392\times 10^{-9}$ m${}^{2}$ K W${}^{-1}$ lies in the experimentally deduced range [37] (2–4) $\times 10^{-9}$ m${}^{2}$ K W${}^{-1}$. To the best of our knowledge, there are no reports of thermal conductivity or thermal boundary resistance measurements for TMS nanocomposites, either in PSL and in NDSL structure.

## 5. Summary and Conclusions

In summary, from investigations made for planar superlattices (PSLs) and nanodot superlattices (NDSLs) constructed from isotropic conductivity covalent materials Si and Ge, and NDSLs constructed from anisotropic conductivity covalent-van der Waals materials MoS${}_{2}$ and WS${}_{2}$, we have identified four key parameters that control thermal transport in nanocomposites. For a given insert fraction, a larger difference between matrix and insert conductivities results in a larger change in the conductivity of the composite material compared to that of the matrix material. Period size *D*, volume fraction of insertion *f*, and atomic-level quality of interface (leading to IMS) are the three other main parameters that should be tuned to achieve low phonon transport in a nanocomposite A/B fabricated from constituent materials A and B. Regardless of sample size *L*, material chemical composition and nanocomposite structural pattern (i.e., PSL or NDSL) and insert volume fraction *f*, the conductivity decreases as the period size *D* decreases towards the low nm range. With regards to equal layer thickness Si(D/2)/Ge(D/2) PSL, the conductivity takes a minimum value when the period size *D* lies in the range 3–12 nm, depending of course on sample size *L* and interface quality. Reported experimental studies [32,33,35,36,37,38,39] and the present systematic theoretical study point out that with the right choice of sample size, period size, volume insertion fraction, and short-range interface defects in a Si/Ge nanocomposite, it is possible to achieve room-temperature conductivity below the alloy and amorphous limit of around 4 Wm${}^{-1}$K${}^{-1}$. This positively points in the direction of the usefulness of nanocomposites for applications such as thermoelectricity.

## Figures and Tables

**Figure 1 nanomaterials-10-00673-f001:**
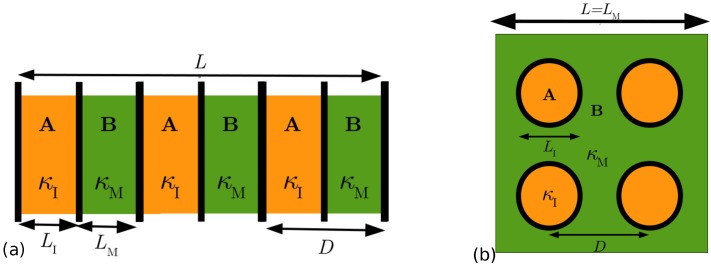
Schematic illustration of: (**a**) a A/B planar superlattice (PSL) structure and (**b**) a nanodot superlattice (NDSL) structure. *L* represents sample length and *D* represents size of a unit cell. We consider A as insert of size $L_{I}$ and conductivity $\kappa_{I}$, and B as matrix of size $L_{M}$ and conductivity $\kappa_{M}$.

**Figure 2 nanomaterials-10-00673-f002:**
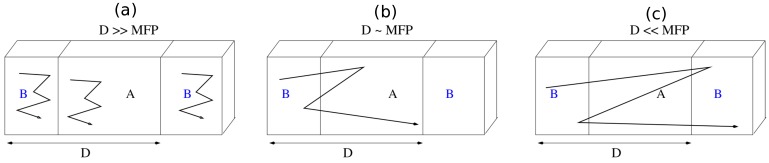
Schematic illustration of relative sizes of repeat period *D* and phonon mean free path (MFP) Λ for a A/B planar superlattice: (**a**) D>>Λ, (**b**) D∼Λ, and (**c**) D<<Λ.

**Figure 3 nanomaterials-10-00673-f003:**
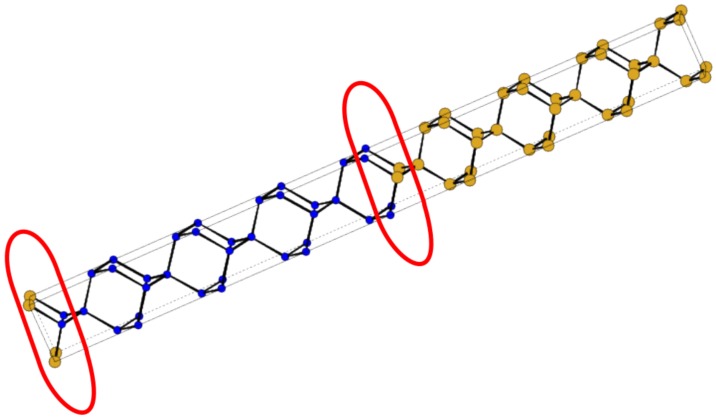
Atomic structure of the Si(D/2)/Ge(D/2)[001] planar superlatiice, with unit cell size D=4.4 nm containing 8 bilayers of Si and 8 bilayers of Ge. Mass smudging has been considered between only the fist interface atomic layers, i.e., in the regions encircled in red.

**Figure 4 nanomaterials-10-00673-f004:**
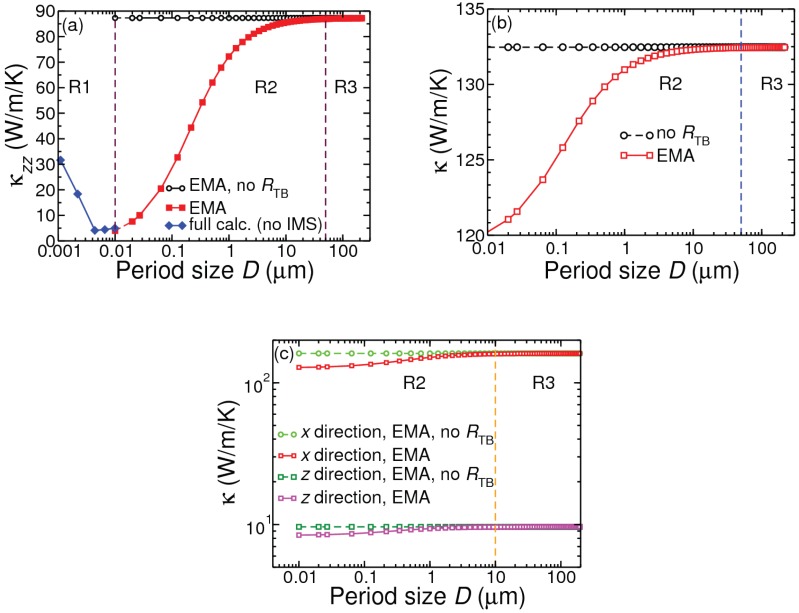
Period size dependence of cross-plane room-temperature thermal conductivity for: (**a**) Si(*D*/2)Ge(*D*/2) PSL; (**b**) NDSL (Ge insert of volume fraction f=0.125 in Si matrix); and (**c**) TMD NDSL (f=0.125 of WS${}_{2}$ inserted in MoS${}_{2}$). Sample size *L* is 1 mm the Si/Ge composites in (**a**,**b**), and 100 mm for the MoS${}_{2}$/WS${}_{2}$ composite in (**c**). Regions R1, R2 and R3 correspond to D<<Λ, D∼Λ and D>>Λ, respectively, where *D* is repeat period and Λ represents phonon mean free path. Results in panel (**a**) for the Si/Ge PSL are obtained from full-scale DFT-Boltzmann calculation in Region R1, and from the EMA method in Regions R2 and R3. Results in panels (**b**,**c**) are obtained from the EMA method.

**Figure 5 nanomaterials-10-00673-f005:**
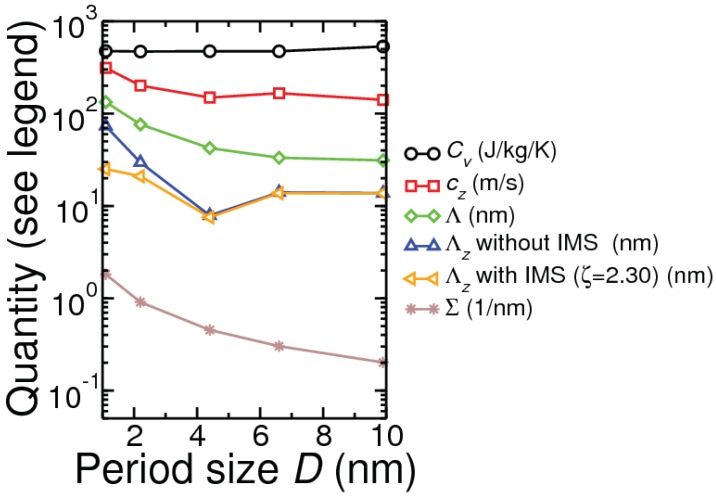
Variation with period size *D*, for Si(D/2)Ge(D/2) PSL at room temperature, of phonon specific heat ($C_{v}$), cross-plane velocity component ($|c_{z}|$), occupation weighted mean free path (Λ), occupation weighted cross-plane mean free path ($\Lambda_{z}$) with sharp interfaces, cross-plane mean free path ($\Lambda_{z}$) with 10% atomic mixing across interface (ζ=2.3), and interface density Σ. Sample size is L=1 mm.

**Figure 6 nanomaterials-10-00673-f006:**
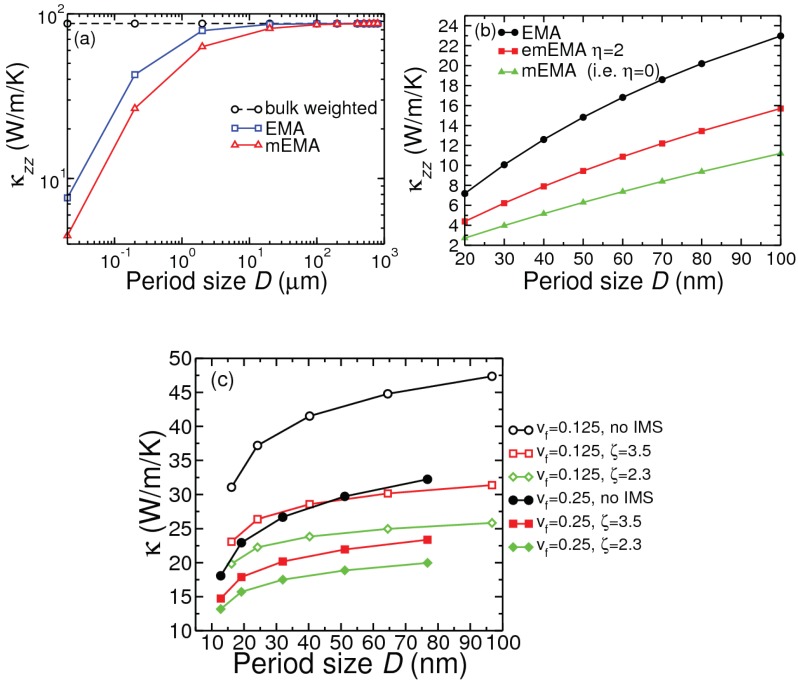
Panels (**a**,**b**): Comparison of period size dependence of room-temperature cross-plane conductivity for Si(D/2)/Ge(D/2) PSL using the EMA, mEMA and emEMA methods. Panel (**c**): Period and volume fraction dependence of κ for Ge ND inserts in Si matrix. Sample size is 1 mm for panel (**a**) and 500 nm for panels (**b**,**c**).

**Figure 7 nanomaterials-10-00673-f007:**
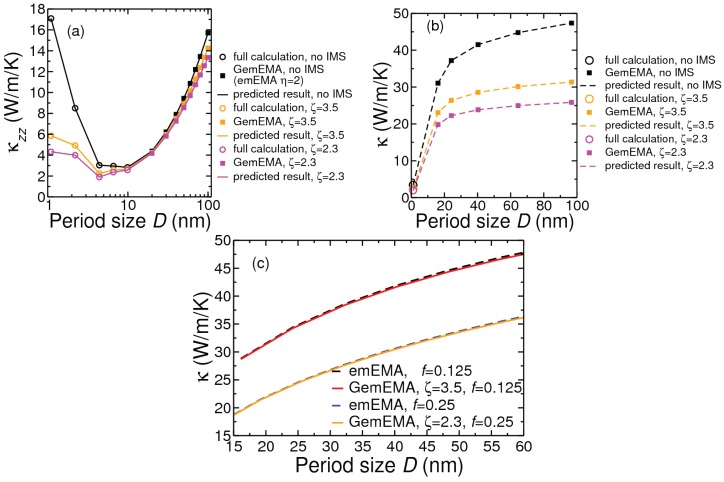
Period size dependence of room-temperature $\kappa_{\mathrm{zz}}$ for sample size 500 nm: (**a**) for Si(D/2)/Ge(D/2) PSL, with various amounts of interface mass smudging (IMS); (**b**) Ge/Si NDSL, with Ge insert fraction of *f* = 0.125; (**c**) MoS${}_{2}$/WS${}_{2}$ NDSL, with WS${}_{2}$ insert fraction of *f* = 0.125. Boundary and interface scattering rates are calculated with the choice of the inhomogeneity factor η=2. In panel (**b**) the shape of the smallest Ge insert was cubic.

**Figure 8 nanomaterials-10-00673-f008:**
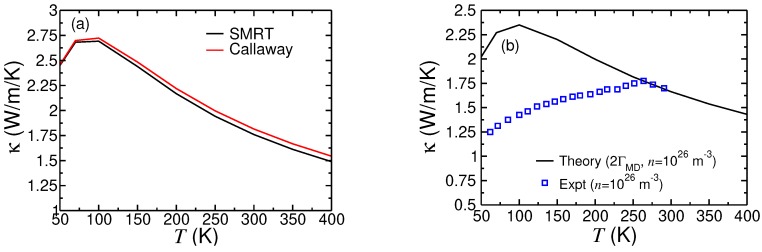
Thermal conductivity of Si(2.2 nm)/Ge(2.2 nm) PSL, with smaple size L=400 nm, n-type doping concentration $n=10^{26}$ m${}^{-3}$, and IMS factor ζ=2.3. (**a**) Results are presented using both SMRT and Callaway theories with L=400 nm. (**b**) Comparison between Callaway theory (this study) and experiment (Ref. [33]), with point mass defect concentration taken as twice of the isotopic mass factor and L=300 nm.

**Table 1 nanomaterials-10-00673-t001:** Table of important quantities and acronyms.

Symbol	Quantity/Acronym
$\kappa_{I}$	Phonon conductivity of insert
$\kappa_{M}$	Phonon conductivity of matrix
$\kappa_{\mathrm{EMA}}$	Effective Medium Approach phonon conductivity
$\kappa_{\mathrm{mEMA}}$	Modified Effective Medium Approach phonon conductivity
$\kappa_{\mathrm{emEMA}}$	Extended modified Effective Medium Approach phonon conductivity
$\kappa_{\mathrm{GemEMA}}$	Generalised and extended modified Effective Medium Approach phonon conductivity
*L*	Sample size
$L_{I}$	Insert length
$L_{M}$	Matrix length
$L_{B}$	Generic effective boundary length
$L_{B,i}$	Effective boundary length of PSL segment *i*
$L_{B,M}$	Effective boundary length of NDSL matrix
$L_{B,I}$	Effective boundary length of NDSL insert
*f*	Concentration of inserts
Σ	Interface density
$R_{\mathrm{TB}}$	Thermal boundary resistance
$W_{\mathrm{IMS}}$	Thermal resistivity due to IMS
*D*	Repeat period size
Λ	Phonon mean free path
ζ	Parameter determining the amount $\exp(-j^{2}\zeta )$ of mass swap in the *j*th interface layer
TMD	Transition metal dichalcogenide
PSL	Planar superlattice
NDSL	Nanodot superlattice
TBR	Thermal boundary resistance
IMS	Interface mass smudging
DFT	Density Functional Theory
SMRT	Single-mode relaxation time
EMA	effective medium approach
mEMA	modified effective medium approach
emEMA	extended modified effective medium approach
GemEMA	generalised and extended modified effective medium approach

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
