# Peer review of "Tunable Thermal Transport Characteristics of Nanocomposites"

_nanomaterials, 2020, doi:10.3390/nano10040673_

Round 1

Reviewer 1 Report

The authors have conducted an extensive and detailed computational study on the thermal transport of nanocomposite materials and identified four key parameters that ideally could be tuned experimentally in order to control thermal conductivity, for potential applications in thermoelectrics for example. This is a valuable contribution that would be a useful guide for the design of nanocomposite thermoelectrics. The downside as the authors point out is the difficulty in experimentally fabricating such nanosized and structurally perfect nanocomposites. With that in mind, it would be beneficial if a wider review of experimental works was performed in order to compare with the theoretical conclusions of this work and specifically to see if these conclusions would hold and to what extent for the more realistic cases of nanocomposites and what is likely to cause the most significant discrepancy between theory and experiment.

Author Response

We thank the reviewer for judging this work a valuable contribution that would be a useful guide for the design of nanocomposite thermoelectrics. We agree with the reviewer that it would be beneficial to provide a discussion on the most significant discrepancy between theory and experiment. We have attempted to do that in Section 4.1.

We have used cyan colour font in the revised manuscript to respond to this reviewer’s comments/suggestions.

Reviewer 2 Report

The author theoretically investigate the thermal conductivity of nanocomposites composed of Si/Ge and MoS2/WS2. Control of thermal conductivities of the materials by designing nanostructures are very important to fabricate thermoelectric materials and the author develop the systematic method to theoretically calculate the thermal conductivities by changing the structural parameters of nanocomposites, which is interesting approach. However, this manuscript is mainly focused on the physics and simulation and I am not sure whether this manuscript has attracted to the general readers in nano-community. And can the author include all schematics of the structures that the authors had used for the simulation? This will  be helpful for the readers to understand what types of structures are simulated. In addition, it would be helpful if the author theoretically calculate a couple of experimental structures, even collected by the others in the literature, using a model the author proposes and compare experimental and calculated results.     

Author Response

We thank the reviewer for their appreciation of our interesting approach of developing a systematic theoretical investigation of the key physical parameters of thermoelectric nanocomposites to control their thermal conductivity. Following reviewer’s suggestion, we have included two new figures (Figs 1 and 2 in the revised manuscript) to clarify the types of structures simulated in our investigation. Also, following reviewer’s suggestion, we have simulated the experimental structure fabricated by Dresselhaus’s group (Ref 33) and provided a reasonably detailed comparison of theoretical prediction and experimental measurements in the temperature range 50-300 K. To the best of our knowledge Ref 33 provides the best fabricated and measured nanocomposite structure in the form of equal-layer Si/Ge planar superlattices. Section 4.1 provides the theory-experiment comparison and a plausible explanation for differences between our predicted results and the measured values (Fig 8b in the revised manuscript).

We have used cyan colour font in the revised manuscript to respond to this reviewer’s comments/suggestions.

Round 2

Reviewer 2 Report

The author has replied all comments. Publication is recommended.